# Spectral Synthesis for Satellite-to-Satellite Translation

## Abstract

Earth observing satellites carrying multi-spectral sensors are widely used to monitor the physical and biological states of the atmosphere, land, and oceans. These satellites have different vantage points above the Earth and different spectral imaging bands resulting in inconsistent imagery from one to another. This presents challenges in building downstream applications. What if we could generate synthetic bands for existing satellites from the union of all domains? We tackle the problem of generating synthetic spectral imagery for multispectral sensors as an unsupervised image-to-image translation problem with partial labels and introduce a novel shared spectral reconstruction loss. Simulated experiments performed by dropping one or more spectral bands show that cross-domain reconstruction outperforms measurements obtained from a second vantage point. On a downstream cloud detection task, we show that generating synthetic bands with our model improves segmentation performance beyond our baseline. Our proposed approach enables synchronization of multispectral data and provides a basis for more homogeneous remote sensing datasets.

## 1 Introduction

Climate change and related environmental issues - including the loss of biodiversity and extreme weather - are listed by the World Economic Forum as the most important risks to our planet (7). Monitoring the Earth is critical to mitigating these risks, understanding the effects, and making future predictions (38). Multi- and hyper-spectral satellite-based remote sensing enables global observation of the Earth, allowing scientists to study large-scale system dynamics and inform general circulation models (26). In weather forecasts satellite data initializes the atmospheric state for future predictions. On longer time scales, these data are used to measure the effects of climate change such as land-cover variations, temperature trends, solar radiation levels, and the rate of snow/ice melt. In the coming decades, increased investments from the public and private sectors in satellite-based observations will continue to improve global monitoring, as highlighted in NASA's decadal survey (25).

Satellites are designed based on specifications for a given set of applications with fiscal, technological, and physical constraints which limit their temporal, spatial, and spectral resolutions. Geostationary (GEO) satellites rotate with the Earth to stay over a constant position above the equator at a high elevation of 35,786km. This position enables GEO satellites with on-board multi-spectral imagers to take continuous and high-temporal snapshots over large spatial regions and are ideal for monitoring diurnal and fast moving events. Spectral bands measure brightness and radiance intensities of the electromagnetic spectrum at a specified center wavelength and bandwidth. Bands are selected to satisfy defined variables of interest constrained by technological cost and accuracy. Applications of GEO sensors include atmospheric winds measurement (35), tropical cyclone tracking (36), wildfire monitoring (41), and short-term forecasting (24). Multiple GEO satellites are needed to generate global high-temporal resolution datasets to better monitor these events around the world. However, variations in resolutions, sensor uncertainties, and temporal life spans leads to a set of separate datasets which are not consistent, making this process very challenging (26). Developing consistent and homogeneous global datasets would relieve many of these challenges.

---

Supplementary Material: `https://github.com/anonymous-ai-for-earth/satellite-to-satellite-translation`

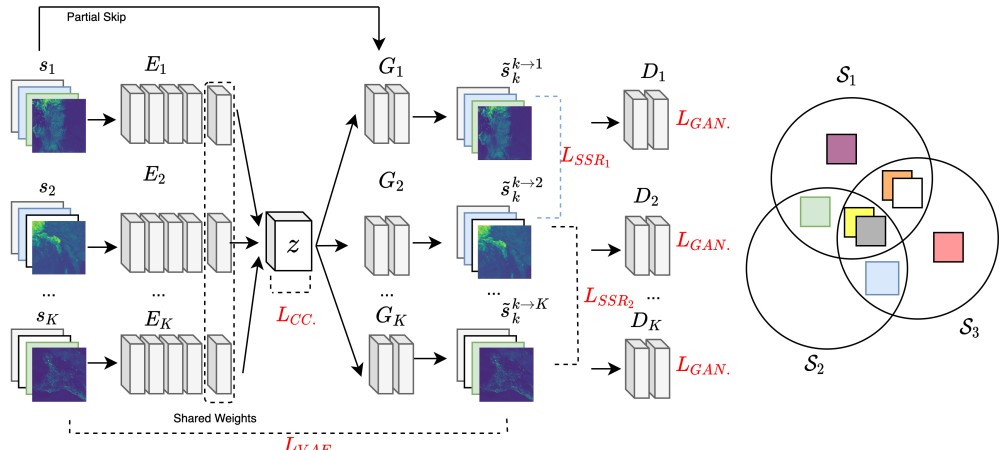

Figure 1: (Left) Network architecture for $K = 3$ satellites. Encoders ($E_k$), decoders ($G_k$), and discriminators ($D_k$) are networks with residual blocks. Losses terms are highlighted in *red*. (Right) Venn diagram shows how spectral bands can overlap between pairs and multiple satellites.

The current generation of GEO satellites are no exception. The GOES-16/17 satellites operated by NASA/NOAA (cost: \$11 billion) have a set of 16 imaging bands covering the visible, near-, and thermal-infrared spectral range (29). The Himawari-8 satellite operated by the Japanese Space Agency (cost: \$800 million) similarly has 16 bands but swaps a NIR ($1.38\mu m$) band for a green channel ($0.51\mu m$), enabling the construction of true color images (3). The $1.38\mu m$ band is ideal for measuring Cirrus clouds, composed of ice particles in the upper troposphere, a major contributor to regulating the Earth's climate that is not yet well understood (21; 10). Without capturing this band, directly observing Cirrus clouds over Japan, East Asia, and Western Pacific region from Himawari-8 is not possible. Synthetic observations via virtual spectral sensors could be a low-cost solution to improving coverage availability and consistency with current satellites.

We present an approach to generate synthetic spectral channels from a multi-domain unpaired satellite dataset. We treat satellites with either dissimilar spectral coverage or varying vantage points as separate spectral sets. In this way, the problem closely resembles that of colorization (40) and image-to-image translation tasks (22; 42; 9) in the case where paired images are not available but with the added complexity of a large number of spectral bands. We use a combination of variational autoencoder (VAE) and generative adverserial network (GAN) (8) architectures adapted to our problem to model a shared latent space, as in unsupervised image-to-image translation(22). Generating synthetic bands is an under-constrained problem that paired with an adverserial loss in high dimensions promotes overfitting. Our approach mitigates these challenges by leveraging a weak supervision signal based on partial overlap in spectral bands between domains. By including a reconstruction loss on overlapping spectral bands between domain pairs we can substantially improve spectral band synthesis.

To summarize our contributions, we 1) introduce a *shared spectral reconstruction loss* to a VAE-GAN architecture for synthetic band generation; 2) test our methodology on real-world scenarios; 3) present and release a test dataset of four hemispheric snapshots from three publicly available geostationary satellites for future research. In the following sections, we will introduce related work in remote sensing and image-to-image translation, describe the architecture, and review experiments. Lastly, we will discuss the implications on this work and conclude with future directions.

## 2 BACKGROUND

**Remote Sensing.** Current generation GEO satellites observe 16 spectral bands over large regions every 10-15 minutes at a 0.5-2km resolution. At a sub-optimal 2km, this produces full-disk images of size 5,424×5,424×16 which causes storage constraints while being computationally expensive to process. Physical and statistical models are used to convert these images into more easily interpreted variables such as precipitation, cloud cover, and surface temperature (30). Multiple GEO satellites, currently in orbit, extend the spatial ranges to actively monitoring larger regions. However, differences

in spectral bands and sensor uncertainties/biases present challenges to commonly used sensor specific models, especially existing downstream models do not generalize well to missing spectral information.

Neural models have long been applied to process remote sensing data and generate downstream products. Hsu et al. (12) presented some of the first work that showed neural networks (NNs) could generate accurate and high-resolution precipitation products from satellite observations. In recent years, convolutional neural networks (CNNs) have been found to further improve this task (27). Similarly, CNNs have successfully been applied to poverty mapping (15), super-resolution (18), subpixel classification (20), model emulation (6), and land-cover classification (2), all from low-level satellite products. In terms of spectral synthesis, few studies have explored reconstruction of hyperspectral bands from RGB bands with supervised approaches (32; 1). While many of these problems are within the class of image-to-image translation, they generally assume labels are widely available and focus on individual sensors. To the best of our knowledge, no studies have developed approaches to synthesize spectral information by learning across satellites in the unsupervised setting.

**Image-to-Image Translation.** Many problems can be defined as an image-to-image translation task including super-resolution, style transfer, and colorization. Approaches to image-to-image translation have been developed for both supervised and unsupervised settings to map images from one domain to another. In the supervised setting, image pairs are available to learn a direct mapping from one to the other. Generative adverserial networks have been shown to be highly successful at this task (14; 33). Numerous unsupervised learning methods have been developed for the common case of large unpaired datasets (22; 42; 39; 23). CycleGAN, for instance, proposed an approach to directly map from one domain to another and back by incorporating a cycle consistency loss with a GAN (42). UNIT (22) proposed a probabilistic approach that uses an intermediate latent space between domains with a Variational Autoencoder (VAE) (16) and GANs (8). In contrast to prior work on image-to-image translation our scenario specifically requires spectral translation and across multiple domains. Rather than translating between relatively low-dimensional RGB images and segmentation maps, as is found in traditional multimodal image-to-image translation (43; 4; 13), satellite imagery contains tens to hundreds of spectral bands. Domain adaptation is another area of active research which also considers the case of effectiveness in unseen environments with cycle consistency and domain invariant (11; 5). (17) using a shared content loss to translate between RGB image styles. (28) presented an application of image-to-image translation for 4-band Sentinel-4 images between different times of day. Our approach is based on the proven fundamental techniques of learning a shared latent space using cycle consistency and adverserial losses extended in the spectral dimension. We also use the prior understanding of spatial consistency between domains to implement a partial skip connection.

## 3 APPROACH

VAEs and GANs are effective for image-to-image translation where pairs of images are not available (22). This is the case with for satellites with no space-time overlap. However, as in (22), a shared latent variable $z$ can be used to approximate the joint distribution from marginals. An adverserial loss applied to cross reconstructions satisfies the shared latent space assumption but is under-constrained for high-dimensional, multi-spectral images. We shall observe that this leads to large errors in our task. To address this, we introduce a shared spectral reconstruction loss and skip connection to effectively generate synthetic spectral bands (see Fig. 1), the result is a 50-80% reduction in mean absolute error.

In the spectral domain, we consider the case of $K$ satellites, $\mathcal{S} = \{\mathcal{S}_1, \mathcal{S}_2, ..., \mathcal{S}_K\}$, such that $S_k \in \mathbb{R}^{H \times W \times B_k}$ is a set of $B_k$ spectral bands with height $H$ and width $W$, illustrated as a Venn diagram in Fig. 1. The union of all sets, $\cup_{i=1}^K \mathcal{S}_k$, represents the complete set of spectral channels in the data. We denote the intersection of two spectral sets as overlapping bands. Our goal is to generate synthetic bands will where $\mathcal{S}_i \cap \mathcal{S}_j^c \neq \emptyset$ for $\forall (i, j)$ where $^c$ denotes the complement. A shared latent variable $z$ is modeled with a Gaussian prior to learn a general representation for mapping between sets such that the assumptions of shared spectral reconstruction, weight sharing, cycle consistency, and cross-domain adverserial losses are satisfied.

**VAE-GAN.** For a given spectral set $k$, we define encoder-generator pairs $\{E_k, G_k\}$ such that $q(z_k|s_k) = \mathcal{N}(E_k(s_k), I)$ and $\hat{s}_k^{k \to k} = G_k(z_k \sim q_k(z_k|s_k))$ for a $s_k \in \mathcal{S}_k$. For any set $j$, $\hat{s}_k^{k \to j}$ corresponds to reconstruction from set $k$ to $j$. The set of encoders $\{E_1, E_2, ..., E_k\}$ share their last layer

of weights to constrain the latent space to high-level representations. Using prior $p_\eta(z) \sim \mathcal{N}(0, I)$, the VAE likelihood is defined as:

$$\mathcal{L}_{VAE_k}(E, G) = \lambda_1 \mathbf{KL}(q_k(z_k|s_k)||p_\eta(z)) - \lambda_2 E_{z_k \sim q_k(z_k|s_k)}[\log p_{G_k}(s_k|z_k)]. \qquad (1)$$

Distributions $p_{G_k}$ are modeled as Laplacian distributions and a Gaussian latent space with prior $z \sim \mathcal{N}(0, I)$. GANs are used to enforce realistic spatial/spectral distributions of reconstructed images from the latent space. Discriminator networks $D_1$ to $D_k$ compare observations with cross reconstructions from the latent space.

$$\mathcal{L}_{GAN_k} = \lambda_3 \mathbb{E}_{s_k \sim P_{S_k}}[\log D_k(s_k)] + \lambda_3 \sum_{j \neq k} \mathbb{E}_{z_j \sim P_{S_k}}[\log (1 - D_k(G_k(z_j)))]. \qquad (2)$$

**Cycle Consistency.** VAE and GAN losses are under-constrained and do not satisfy the shared latent space constraint alone. As in (22), a cycle consistent loss is used such that $s_k = F_{j \rightarrow k}(F_{k \rightarrow j}(s_k))$ for all satellite pairs $(j, k)$ and where $F_{k \rightarrow j}(s_k) = G_j(E_k(s_k))$. The loss between $s_k$ and cycled reconstruction $\hat{s}_k^{k \rightarrow j \rightarrow k}$ is written as:

$$\mathcal{L}_{CC_{k \rightarrow j}}(E_k, E_j, G_k, G_j) = \lambda_4 \mathbf{KL}(q_k(z_k|s_k)||p_\eta(z)) + \lambda_4 \mathbf{KL}(q_j(z_j|s_k^{k \rightarrow j})||p_\eta(z)) \\ - \lambda_5 \mathbf{E}_{z_j \sim q_j(z_j|s_k^{k \rightarrow j})}[\log p(G_k(s_k|z_j))] \qquad (3)$$

With multiple domains, each domain should cycle through every other domain. The cycle-consistency loss for each permutation results in a complete cyclical graph. This loss is written as:

$$\mathcal{L}_{CC_k} = \sum_{k \neq j} \mathcal{L}_{CC_{k \rightarrow j}}(E_k, E_j, G_k, G_j) \qquad (4)$$

**Shared Spectral Reconstruction Loss.** Adverserial losses can be easily fooled with increased dimensions. To help avoid this we introduce an additional loss, $\mathcal{L}_{SSR_k}$. In this problem, if the intersection of spectral channels $\mathcal{S}_{k,j} = \mathcal{S}_k \cap \mathcal{S}_j$ between domains is not empty then the difference between $p(s_k^{k \rightarrow k}|z_k)$ and $p(s_k^{k \rightarrow k}|z_k)$ can be minimized with KL divergence:

$$\mathcal{L}_{SSR_k} = \lambda_6 \sum_{j \neq k} \mathbf{KL}(p(\tilde{s}_k^{k \rightarrow k}|z_k)||p(\tilde{s}_k^{k \rightarrow j}|z_k)) \qquad (5)$$

where $\tilde{s}_k \in \mathcal{S}_{k,j}$. The $SSR$ loss encourages decoders to reconstruct identical spectral wavelengths with similar distributions while still synthesizing dissimilar bands. In this scenario, partial constraints are placed between domains and allows sampling of unobserved spectra from the shared latent space. By decreasing $\lambda_6$ the bias between bands will be relaxed which may reduce the effect of more uncertain domains.

**Total Loss** The likelihood is maximized by optimizing the GAN mini-max problem such that the generator aims to fool the discriminator, alternating updates between $(E, G)$ and $(G, D)$.

$$\mathcal{L} = \min_{E,G} \max_D \sum_{k=1}^{K} \left[ \mathcal{L}_{VAE_k} + \mathcal{L}_{CC_k} + \mathcal{L}_{GAN_k} + \mathcal{L}_{SSR_k} \right] \qquad (6)$$

The hyper-parameters used correspond to those in (22) and set as $\lambda_1 = 1, \lambda_2 = 0.01, \lambda_3 = 1, \lambda_4 = 1, \lambda_5 = 0.01,$ and $\lambda_6 = 0.1$. Adam optimization is used to train the networks for 200,000 steps with a batch size of 8 with parameters $\beta_1 = 0.5, \beta_2 = 0.999$ and learning rate $1e - 5$. The reader can find detailed information in the supplementary material. Below we show the steps for generating a new band.

---

**Algorithm 1:** Generate a synthetic band by translating from one satellite to another

---

**Result:** Synthetic spectral band
Image $s_k$ from satellite k;
Encode to latent space $z = E_k(s_k)$;
Decode to other satellite $\tilde{s}_j = G_j(z)$;
Select synthetic band from $\tilde{s}_j$;

---

**Data.** Three geostationary satellite imagery datasets, GOES-16 (G16), GOES-17 (G17), and Himawari-8 (H8) are used in our experiments. Each satellite captures hemispheric (full-disk)

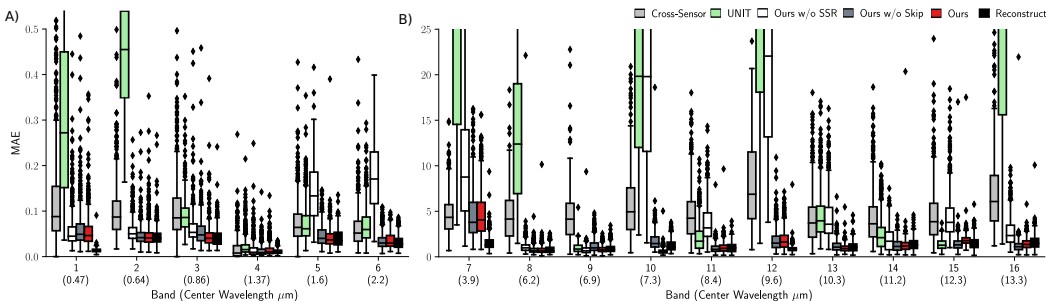

Figure 2: Mean absolute error (MAE) of a substitute sensor (GT observations from a separate satellite), synthetic sensor without SSR, synthetic sensor without skip, synthetic sensor, and reconstructed sensor (reconstructed images from a full model with access to all bands) for each band. The wavelength of each band is shown below the band number. A) include visible and near infra-red (no physical units) and B) includes thermal infrared measured in Kelvin.

snapshots from a constant vantage point over time but of different regions. Examples shown in supplement. Images contain 16 bands (channels) in the visible, near-infrared, and thermal spectrum at 0.5-2km spatial resolution. G16 and G17 have identical specifications viewing the east and west regions of North America and include two visible (blue, red), four near-infrared (including cirrus), and ten thermal infrared bands. H8 has 15 overlapping bands with G16/G17 viewing the Pacific Ocean and East Asia, this ensures similar information content. H8 captures three visible (blue, green, red), three near-infrared (missing cirrus), and the same ten thermal infrared bands as G16/G17. Thus, the G16 and G17 bands all overlap, cirrus ($1.37\mu$m) exists in G16 and G17 but is not in H8 and green ($0.51\mu$m) exists in H8 but not in G16 or G17. These differences cause difficulties when applying models relying on green or cirrus bands across satellite sets.

G16 observes the North, Central and South Americas, capturing a good distribution of land and ocean. G17 observes the Pacific Ocean as well as most of North and Central America. However, G17 has known problems with its thermal cooling system causing the near-infrared and thermal-infrared channels to be unusable during periods of high heat and biased throughout (31). This further highlights the gain in replacing low quality bands of G17 with a virtual sensor. Periods of high heat are filtered out of our training and test sets with quality control checks to eliminate temporal periods of known uncertainty. After quality control, considerable space-time overlap between G16 and G17 can be used for testing. H8 observes East Asia, Australia, and the Western Pacific, partially overlapping with G17. Discrepancies are expected between sensors caused by different solar and sensor viewing angles but we are not aware of a more appropriate dataset for evaluation. The data generated by (37) is used which normalized G16, G17, and H8 to a common geo-referenced gridding system in order to facilitate intercomparisons and processed with the Bidirectional Reflectance Distribution Function (BRDF). Bands have resolutions varying from 500m to 2km which we interpolate to a common sub-optimal resolution of 2km. Full-disk images are on a common grid with *tiles* of size $300 \times 300 \times 16$. Training data is generated from the multi-petabyte datasets. We randomly sample images to build a well distributed and large training dataset from years 2018 (G16,H8) and 2019 (G17) which totaled 359GB of data. Each tile is split into $64 \times 64 \times 16$ non-overlapping patches for training, generating millions of samples.

A test set including 500 random tiles from 25 days in February 2019 from overlapping G16 and G17 observations. The random set of tiles assures a range of solar angles, system patterns, and land cover types. Similarly, four tiles of data from G17 and H8 on January 2, 2019 at 04:00UTC are selected to evaluate synthetically generated green and cirrus bands (spatial overlap of G17/H8 is mostly ocean). This dataset will be made publicly available consisting of tiles from each satellite.

## 4 EXPERIMENTS AND DISCUSSION

In this section we present a set of experiments to explore the properties of our approach by testing which bands can be robustly synthesized, how many bands can be generated, how effectively the proposed loss performs, and the ability to perform downstream tasks.

**Cross Satellite Band Synthesis.** Our experiments start with testing how well each spectral band can be synthesized. To do this we remove individual bands from one satellite (G16) during training, synthesize these bands and compare with the ground-truth observations. We use the full set of bands from the other two satellites during training. This approach is applied on G16 such that each model takes 15 bands of G16 and 16 of G17 and H8.

Three comparisons are used to help put the accuracy of synthesized bands for G16 into context. *Ours* refers to bands generated using our proposed approach with both SSR loss and skip connection. *Ours w/o SSR* refers to bands generated with our proposed approach without the shared spectral reconstruction loss. *Ours w/o Skip* refers to bands generated with our proposed approach without the skip connection between the input and generator. *UNIT* refers to the unsupervised image-to-image translation baseline as presented in (22) and is equivalent to ours without SSR and skip connection. *Sensor* refers to the performance if we simply use overlapping observations from another satellite (G17), this acts as our lower bound in performance and is actually the status-quo (essentially substituting the missing band with images from the same band but from another satellite, which as we shall see is a sub-optimal solution). *Reconstruction* refers to the images reconstructed from a full model trained on all satellites with no missing bands, this acts as our upper bound in performance. Each of these signals are computed on our test set of 500 overlapping overlapping tiles from G16/G17. Fig. 2 shows the mean absolute error (MAE) for each condition. Table 1 shows the average MAE for VIS/NIR and TIR of each method.

Table 1: The MAE for VIS/NIR and TIR bands by method.

| Method | VIS/NIR | TIR |
|---|---|---|
| Cross-Sensor | .186 | 4.45 |
| UNIT | .183 | 12.64 |
| Ours w/o SSR | .101 | 7.84 |
| Ours w/o Skip | .049 | 1.68 |
| Ours | **.048** | **1.48** |
| Reconstruct | .036 | 1.14 |

The MAE in the sensor condition is substantial and largely caused by clouds/aerosols in the vertical direction (see gif in supplementary material). On the other side, synthetically generating bands using our approach substantially reduces error by over 60% compared to both this baseline and UNIT(see Table 1). Similarly, synthesized bands also improve upon the view from G17 even though during training they did not see examples of the corresponding band from G16. Overall, the reconstructed and synthesized images have similar signal-to-noise ratios. Ablation experiments removing the shared spectral reconstruction loss and the skip connection show their effectiveness. $SSR$ is critical to learning a robust latent space and the skip connection improves both VIS/NIR and TIR predictions. We observe that without introducing the SSR loss, performance is even worse than the sensor baseline. From this we learn that applying an existing image-to-image translation model (22) to our task, without adaptation, performs poorly. We find that band 7, the shortwave infrared band ($3.9\mu$m), is particularly difficult to synthesize with MAE significantly above that of the full reconstruction. This result suggests that the shortwave infrared band captures information which cannot be inferred from the others. Notice how the wavelength gap between bands 7 and 8 is relatively large ($2.3\mu$m), this may explain why the performance is poor. In the future, a similar analysis could be used to inform future satellite design configurations.

We show qualitative examples of generating synthetic bands in Figs. 3b and 4. Two images are shown in Fig. 3b including a false color image, composed of near-infrared, red, and blue bands, and a true color image from a synthetically generated green band and real red and blue bands. This process is applied to Himawari-8 to generate a cirrus band (shown in Fig. 4). While there may be challenges in synthetically generating all bands, most can be reconstructed with a high signal-to-noise ratio and this suggests that our approach could be used to make software updates to current satellite datasets.

**Synthesizing Multiple Bands.** Generating synthetic channels from satellites with a limited number of spectral bands could be of significant value for long-term analysis. For example, older generation satellites often have fewer channels and could provide greater utility in downstream tasks if it was possible to generate images in additional frequency bands. Therefore, we set up an experiment to test how many additional bands can be synthesized reliably and how many initial bands are required. A set of synthesis models were trained on G16, removing bands one by one until just one band was left and while keeping all 16 G17 and H8 bands. For simplicity, and to reduce computation, we dropped bands in a fixed order: 9,4,13,2,15,12,6,3,10,8,14,5,11,7,16. In the most extreme case we use visible band 1 and attempt to synthesize the remaining 15. As above, results are computed on the test set of 59 overlapping G16 and G17 tiles. The results presented in Fig. 3a (left) show how the number of available input bands effects the MAE for VIS/NIR (bands 1-6) and TIR (bands 7-16). As expected, MAE falls more or less monotonically as more bands are given as inputs. When just two bands, 1

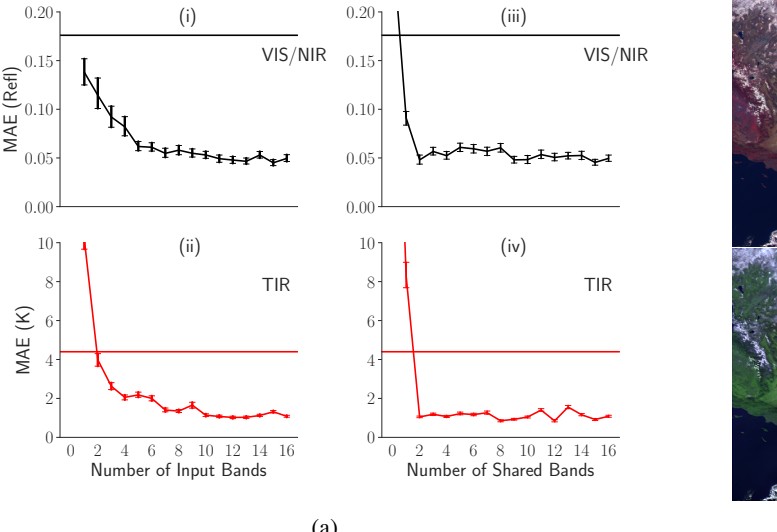 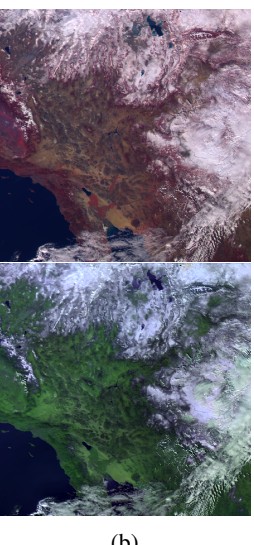

(a)                       (b)

Figure 3: (a) Horizontal lines correspond to the sensor signal and error bars represent 95% confidence intervals of tile MAE. (i-ii) show results of an experiment synthetically generating more and more bands. (iii-iv) show MAE as a function of number of bands shared between spectral sets. (i and iii) correspond to visible/near-infrared and (ii and iv) to thermal-infrared. (b) False (top) and true (bottom) color images generated by GOES-17 on January 1 2019.

(blue) and 16 (TIR), are used as inputs, the synthetic TIR reconstruction of G16 still has lower error than the observed sensor difference between G16 and G17. These results show that few bands are needed to synthesize images that improve upon the status quo. In the TIR range, we find that MAE plateaus after 3-4 bands are used as inputs. These results suggest that the information content in a subset of bands may be sufficient for many applications. However, we should be prepared that some bands may contain specific information useful for monitoring rare events. Overall, these results show that a good proportion of bands can be synthesized remarkably well.

Table 2: The MAE for VIS/NIR and TIR bands by $\lambda_6$.

| $\lambda_6$ | VIS/NIR | TIR |
|---|---|---|
| 0.01 | .068 | 5.03 |
| 0.10 | .048 | 1.48 |
| 1.0 | .042 | 1.34 |
| 10.0 | .049 | 1.28 |

**Sharing Spectral Losses.** The effectiveness of the shared spectral reconstruction loss is tested by gradually increasing the number of shared bands included in the loss one by one. Mathematically, this corresponds to the number of bands included in the set $\mathcal{S}_{k,j}$. In all runs, 16 bands of G16, G17, and H8 are used even if ignored by the $SSR$ loss. Fig. 3a(iii,iv) shows the effect of adding shared bands during training leads to a dramatic decrease in MAE. Corresponding cross-sensor signals are shown as horizontal lines. In this setting, we find using the $SSR$ loss is critical to learning this model. Sharing two spectral bands in the loss function improves the signal and is almost all that is needed for accurate reconstruction. This further reinforces our insight above that a large amount of the information is captured in just a few spectral bands. In Table 2 we further explore the $SSR$ loss by testing a range of values for $\lambda_6$ from 0.01 to 10. Our results suggest that increasing $SSR$ weighting factor improves performance on the test set.

**Synthesizing Cirrus for Himawari-8.** As discussed above, the cirrus band (1.38 $\mu$m) monitors ice particles in the upper troposphere which regulate the climate, and H8 is missing this band. These ice particles are often seen as thin clouds high in the atmosphere which may be viewed in the visible range, along with other clouds. To generate a synthetic cirrus band, an H8 observation is translated to G17. In Fig. 4 we show five images where G17 and H8 have space time overlap. G17 observations of false color and cirrus bands present a baseline. The observed H8 true color depicts the same scene and a corresponding synthetic cirrus band. This scene consists of clouds of multiple types and atmospheric heights on January 2, 04:00UTC. Cirrus clouds are found high in the atmosphere and are seen as thin or wispy (see lower right portion of the images). Comparing images 4b and 4d shows the similarity between synthetic bands and observations. Lower-level clouds, which can be seen on the lower right of the images, are ignored by both the observed and synthetic cirrus bands. Fig. 4e also shows the corresponding latent space from H8 where we highlight the feature maps corresponding to

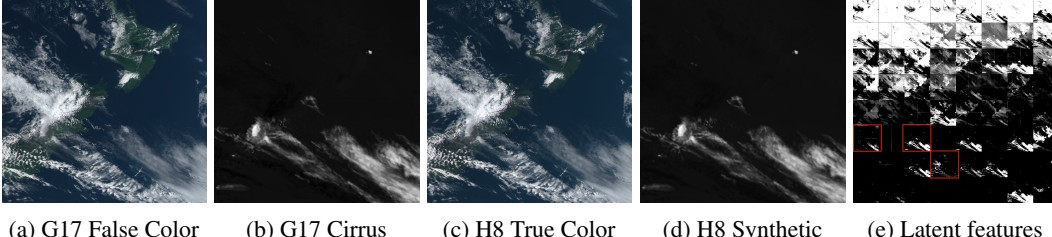

(a) G17 False Color    (b) G17 Cirrus    (c) H8 True Color    (d) H8 Synthetic    (e) Latent features

Figure 4: Synthetically sensing cirrus from Himawari-8 using GOES-17. Images are taken from the test set (January 2, 2019 at 4:00UTC). (a) G17 false color image (NIR,R,B) (b) observed cirrus, (c) corresponding true color H8 image and (d) synthetically generated cirrus. Latent features are shown in (e) with red boxes highlighting feature maps capturing cirrus clouds.

the cirrus band. These suggest that our approach has learned to distinguish that clouds of different types are visible from different bands, a particularly important result.

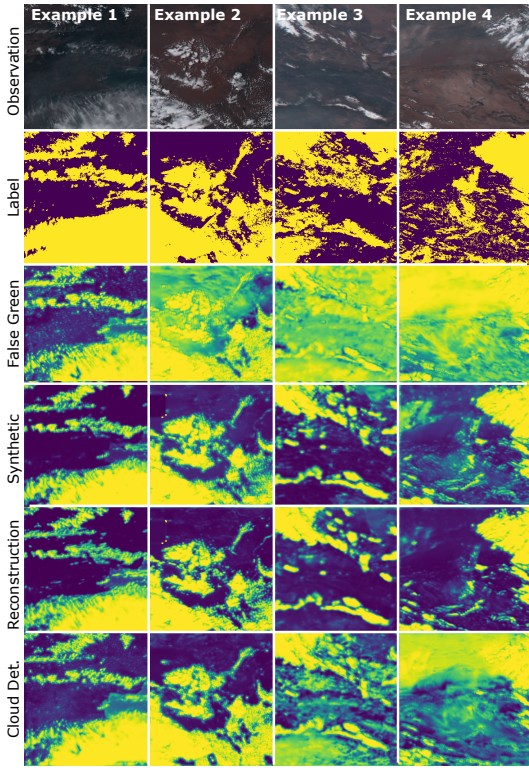

Figure 5: Cloud segmentation of four Himawari-8 observations. From top to bottom, rows show observation and cloud mask labels followed by false color, synthesized, reconstructed, and observed segmentation. Yellow denotes cloud pixels and blue non-cloudy pixels.

**Cloud Detection.** The goal of generating synthetic satellite datasets is to enable us to run downstream applications and models. Cloud/aerosol segmentation is one important task. To demonstrate this we take a learned cloud detection model that is dependent on H8 bands 1-4, *including the green band*. However, the green band is not available from G16/G17 and therefore we cannot directly use the cloud detection model with these data, we need to synthesize the green band first. For background, the cloud detection model was trained using labels for clouds and aerosols from a dataset generated with a physically-based land surface model tuned specifically for H8 (19). With this training dataset, a model can be trained to perform cloud detection (19), specifically we use a Bayesian convolutional neural network as introduced in (34; 6) for the model.

Now we can compare the cloud detection accuracy with and without H8's green band and with a synthetic green band. We take a version of our synthesis model trained to synthesize a green band. Our model is able to translate between H8 views with and without green bands. Quantitatively, we compute the area under the receiver operating characteristic curve (AUC) when comparing labels to segmented probabilities over 100 random samples. On the upper limit, observed and reconstructed AUCs are 0.93 and 0.89, respectively. The AUC falls to 0.73 using a false green band versus 0.88 for the synthetically generated green.

In Fig. 5, we show qualitative examples of emulated cloud detection results. The first row shows the inputs as true color observation with visible clouds and the second shows the physically generated cloud/aerosol labels. As a baseline, a false green is generated as the average of red and blue bands (R,$\frac{R+B}{2}$,B) and used for cloud detection. The fourth row shows the results using our synthetic green band translating between 15-band H8 and 16-band H8. Similarly, the last two rows show results when using reconstructed and observed bands for segmentation, respectively. Segmentation from the false color images often produces unclear results and overestimates cloud cover. In contrast, the

synthetically generated bands produce segmentation maps that look nearly identical to reconstructed and observed examples. In the last column, we show an example of our approach acting as a denoiser to improve cloud detection even beyond the observed cloud segmentation. In sum, results suggest that synthetic data generated from our approach is applicable and may even have the potential to improve downstream tasks.

**Limitations.** While the VAE-GAN architecture performs well overall, it does present some limitations. VAEs aim to explicitly model the data as a multivariate Gaussian and often produces blurry outputs. The GAN counteracts this effect by discriminating between real and generated images. However, there is concern that this reduces data precision and fails to detect rare and anomalous events which may effect scientific applications. Extending our work to use normalizing flows, as in (9), may reduce this limitation.

## 5  CONCLUSION

We have presented an unsupervised learning approach for satellite-to-satellite translation that can be used to synthesize unobserved spectral bands. A novel shared spectral reconstruction loss is presented to further constrain learning and conserve spectral information and a partial skip connection maintains spatial consistency. Experiments with sensors on the GOES-16/17 and Himawari-8 satellites show that synthetic spectral bands can be generated through reconstruction from a shared latent space. For the first time, we are able to generate true color images from GOES-16/17 and the cirrus band from Himawari-8, generating further value from these satellites. Further, a cloud detection model is used to show the applicability of synthetically generated bands for downstream tasks. Future work may consider conditioning the shared latent space with known physical properties and extending to additional tasks.

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

# A APPENDIX

## A.1 TEST IMAGES

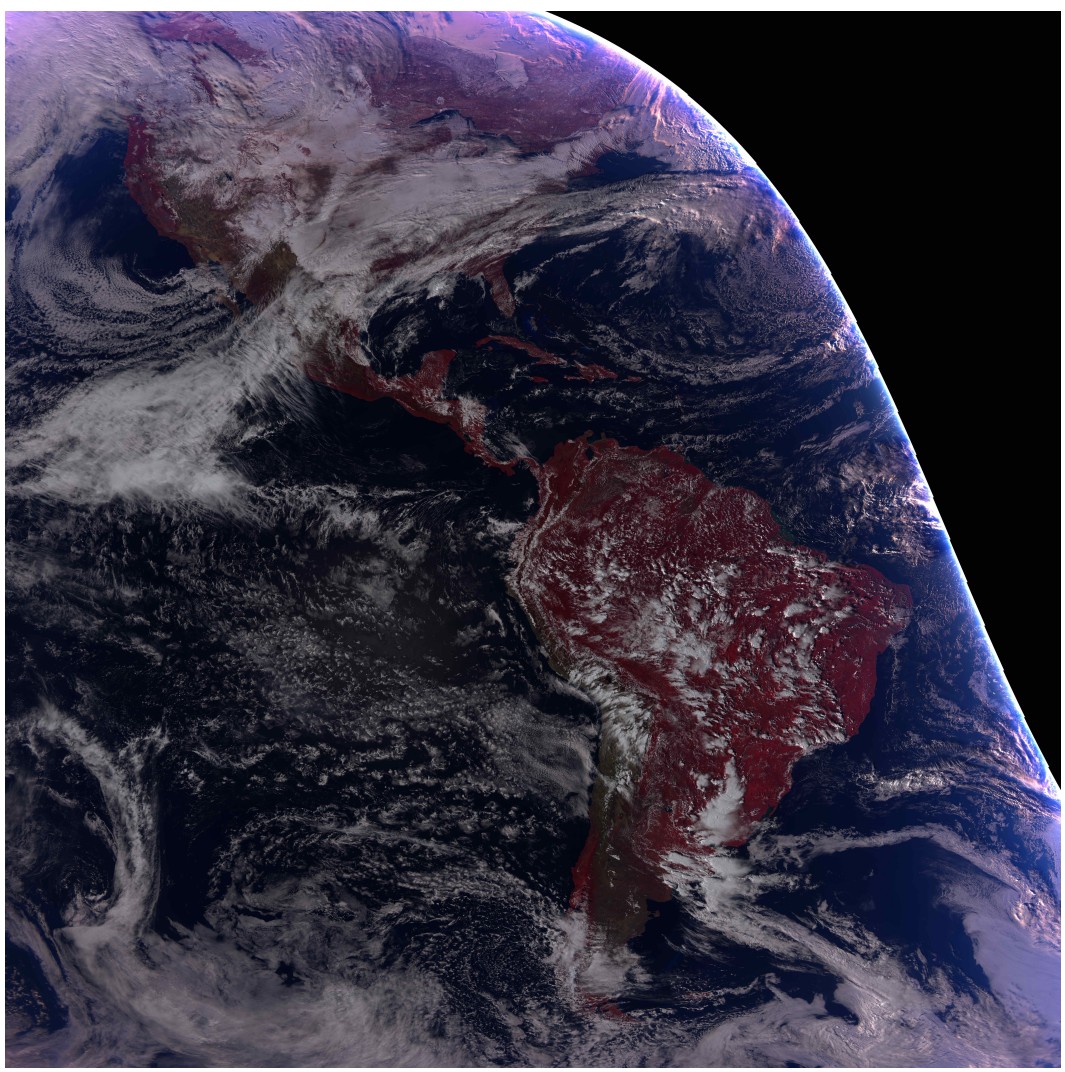

Figure 6: Full Disk Image of GOES-16 at 20:00 UTC on January 1 2019 - False Color (NIR,Red,Blue)

You may include other additional sections here.

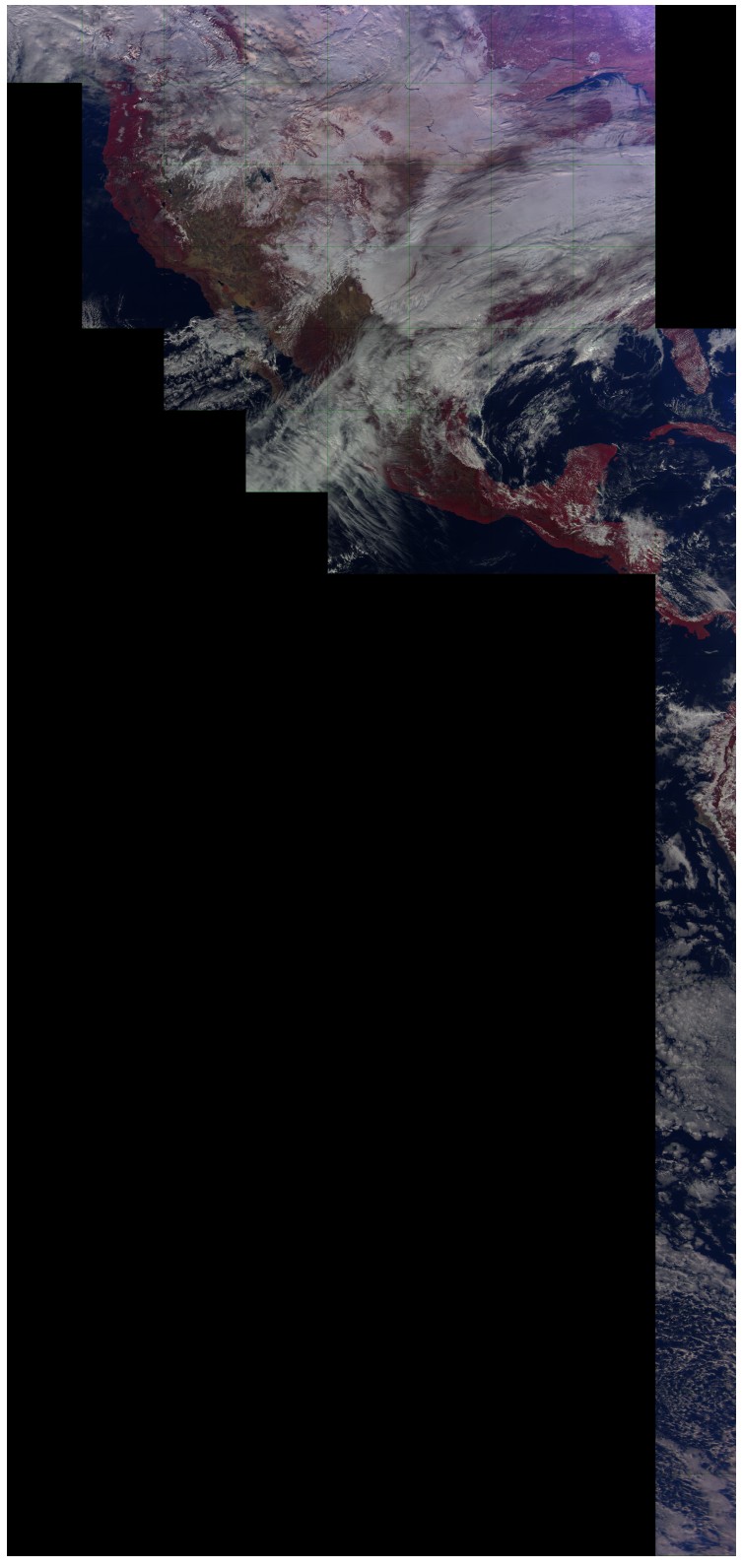

Figure 7: GOES-17 Coverage overlapping GOES-16 at 20:00 UTC on January 1 2019 - False Color (NIR,Red,Blue)

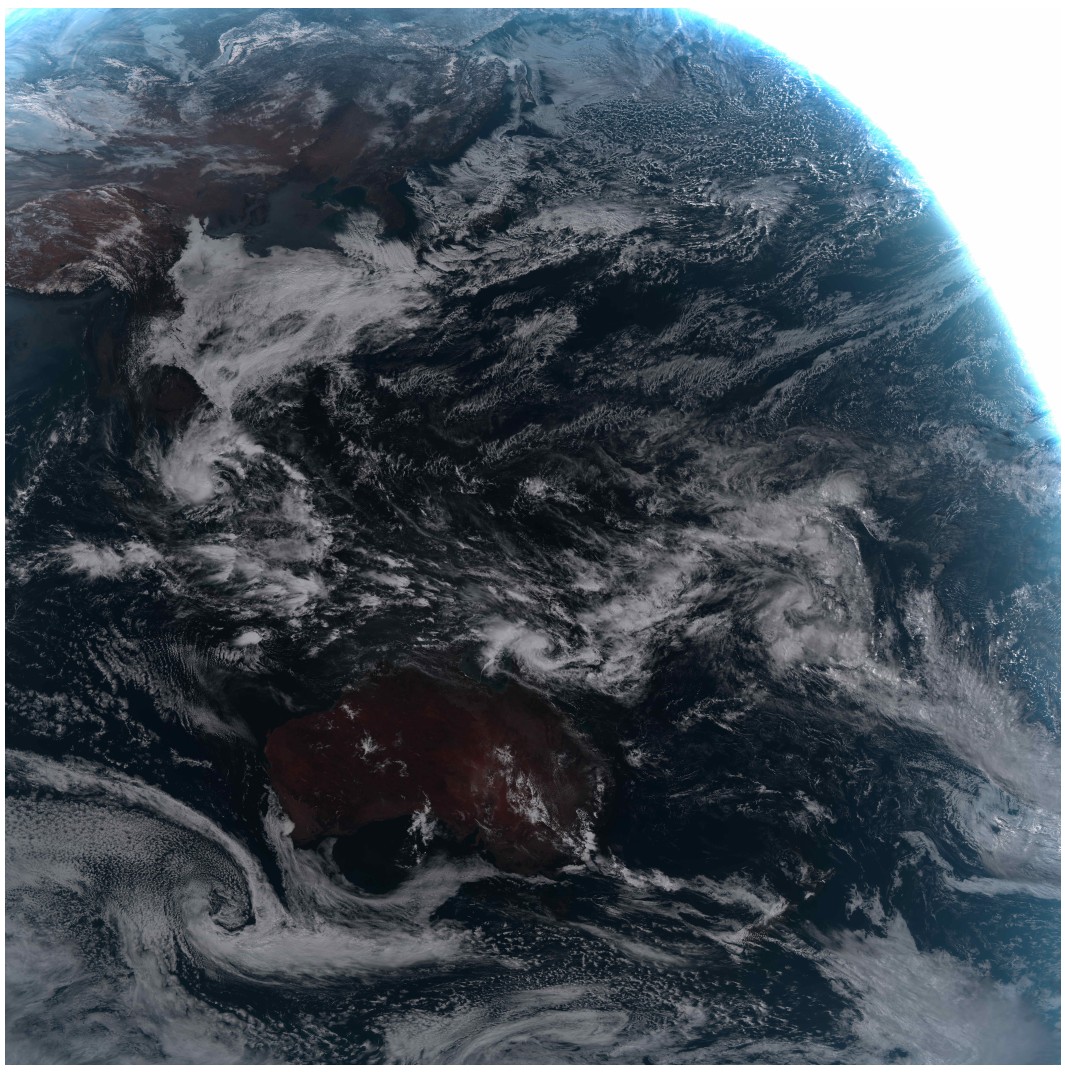

Figure 8: Full Disk Image of Himawari-8 at 04:00 UTC on January 2 2019 - True Color (Red,Green,Blue)

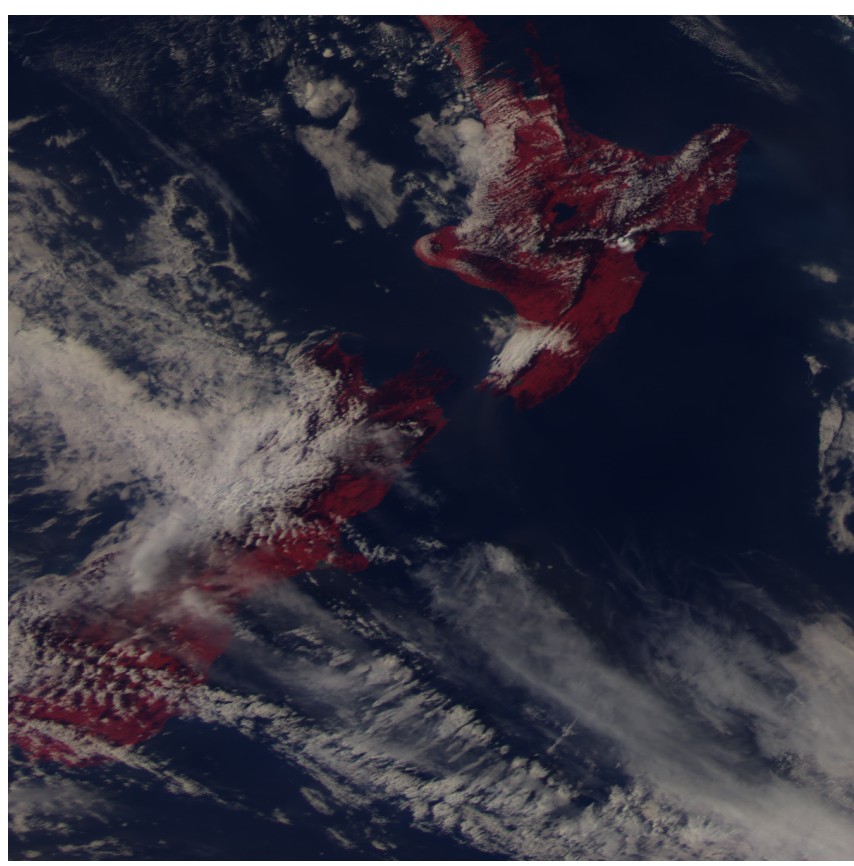

Figure 9: GOES-17 Coverage overlapping Himawari-8 at 04:00 UTC on January 2 2020 - False Color (NIR,Red,Blue)

