# OpenReview forum: "Spectral Synthesis for Satellite-to-Satellite Translation"
_ICLR.cc/2021/Conference — Reject_

### Official Review · AnonReviewer1 · 2020-10-28
**Good application of VAE-GAN, but the contribution is not big**

**Rating:** 5
**Confidence:** 4

**Review:**

This paper tries to generate synthetic unobserved spectral imagery from a set of existing spectral channels. This is an image-to-image translation task, for which VAEs and GANs are effective to address. For this specific satellite band-to-band translation task, authors adopt the VAE-GAN framework (adding a skip connection between the input and generator), with a new added spectral reconstruction loss.  Experiments show the effectiveness of the proposed method.

Pros:

1. As stated by the authors, this paper introduces a shared spectral reconstruction loss to a VAE-GAN architecture for synthetic band generation.
2. Simulated experiments show that the generated banded is possible and effective.
3. Using the generated band, the proposed model improves segmentation performance beyond baselines.
4. A test dataset of four hemispheric snapshots from three publicly available geostationary satellites is thankfully received.


Cons:

I would say this paper has limited contribution, in terms of the novelty of the method. For me, this paper is an application paper. Specifically:

1. I would say among the three contributions (listed before section 2. background), only the first contribution of proposing a new loss makes sense for me as we always need to validate methods on real datasets. For the proposed shared spectral reconstruction loss, it's a simple KL divergence loss (Equation 5) imposed on overlapping channels, specifically tailored for this band-to-band translation task under the VAE-GAN framework. In one word, the contribution is limited.

2. Throughout the experiments section, I cannot find a baseline method from a third-party. The reason could be that no one did this task before, which is good. However, including an off-the-shelf image-to-image translation method as a baseline is necessary. It will help readers to understand the difficulty of this band-to-band translation task.

3. To make the proposed shared spectral reconstruction loss stronger, I would like to see an ablation study concerning how it interacts with other losses in equation 6. Please use a different number of $\lambda_6$ to illustrate.

In summary, I would say this paper demonstrates the success of VAE-GAN on the band-to-band reconstruction task, which is good, but with limited technical novelty.

---

> ### Author Response · Authors · 2020-11-15
> **Response to AnonReviewer1**
>
> We thank the reviewer for noting the importance and effectiveness of our reconstruction task as well as the comments which will improve the paper. The goal of this work is to find a solution to a new task of matching spectral bands across different sensors on a real world problem and benefits from the well understood methodology of VAE-GANs for image-to-image translation.  Our methodology is shown to make critical adaptations to the VAE-GAN architecture to solve this problem. We would argue that these are non-trivial adaptations. We believe this research brings to light a new set of applications and could motivate future methodological developments in representation learning.
>
> Baseline:  As pointed out, we are not aware of studies similar to this for a traditional baseline and believe our work presents a baseline for future research in this area. However, our methodology is based on the UNIT [23] VAE-GAN architecture with the addition of a spectral reconstruction loss and skip connection. UNIT is equivalent to Ours w/o Skip and SSR which the results for which were not originally included in Table 1. We have included the results for UNIT in the revisions as our baseline. Results show that our approach performs 60% better than UNIT on VIS/NIR bands and 90% on TIR bands.  UNIT performs worse than the cross-sensor baseline when considering all the bands. In the text, we more clearly delineate this within the results section.
>
> SSR loss: We agree that better understanding the range of $\lambda_6$ is helpful for readers and optimization. We have run experiments for $\lambda_6$ for values [0.01, 0.1, 1., 10.] and include the results in Table 2 of the revisions. This experiment found that a weight of 0.01 was not effective for learning but 0.1, 1.0 and 10 each performed similarly. This also showed that our results can be further improved by increasing $\lambda_6$ to 1.0. We appreciate your comment on this.

---

### Official Review · AnonReviewer2 · 2020-10-28
**This manuscript is an application of unsupervised VAE-GAN model for satellite-to-satellite translation.  The application is meaningful, whereas the method is weak.**

**Rating:** 6
**Confidence:** 4

**Review:**

This paper applied the unsupervised VAE-GAN model for satellite-to-satellite translation to generate synthetic spectral bands. My general evaluations of this paper are:
pros: The application is meaningful. This paper exploits the possible application of VAE-GAN based image-to-image translation model to the generation of synthetic spectral bands. Of course, it will bring new solutions for the subsequent application of satellite images. From the method perspective, the proposed Shared Spectral Reconstruction Loss (SSRS) has been testified to significantly improve the final spectral band simulation results.
cons: From the model perspective, the only shining point compared to the previous model is the introduction of SSRS loss.  The notations are not clear. In the experiments, the comparison methods are missing. It is hard to distinguish the performance of the proposed model.

More general comments are listed as the following.
There are so many notations in the proposed method introduction,  however, the meaning of notations to the satellite images are not clear. I think most notations are from ref[23], but the most contribution of this paper is the application to satellite images. It may be more meaningful to introduce the notations from the satellite images perspective.

The experimental dataset is complex, but the introduction is insufficient.  It is difficult to make the readers understand the task of the experiment. I suggest to enhance the introduction of the problem and formulate the mathematical model from satellite-to-satellite translation clearly.

In the experiments, it is hard to rate the performance of the proposed model. There are no comparisons to previous state-of-the-art, and I'm not so familiar with this dataset. The authors should pay more attention to make their datasets understood to the readers, including the performance judgment and possible further applications.

---

> ### Author Response · Authors · 2020-11-15
> **Response to AnonReviewer2**
>
> We thank the reviewer for their positive comments, especially noting the importance of the problem and opportunity for new solutions in the application of satellite imagery, which is a critical tool in climate science research. Our hope is that this research opens directions for solving problems throughout the remote sensing processing chain. As we demonstrated, our solution, while adding critical terms to [23], improves predictive performance on the downstream task of cloud detection. This represents the generation of a level 2+ remote sensing product which includes tasks such as monitoring weather, radiative transfer models, land-cover classification, and others.
>
> Baseline and notation: We agree with the reviewer that notation delineating between baselines and our methodology is unclear. After running experiments, we have revised the paper with a new row in Table 1 with the baseline of “UNIT [23]”, which removes the SSR loss and skip connection from our approach.  Our proposed changes to the model result in large improvements over this baseline (60%+ reduction in error). UNIT performs worse than the cross-sensor baseline. We have made updates to the notation (see Section 3) to improve readability and to align symbols more closely with concepts they represent (ie. satellites changed from X -> S, and bands changed from C -> B).
>
> Dataset: Given the increased length allowed in the revisions, we have added further explanation of our dataset in Section 3 (subsection Data). We appreciate the ask for the link to the data, for anonymity it can be accessed here: https://drive.google.com/file/d/1gzLcqWiKPjvzltp2nVZH6uCSIQ0G2h1u/view?usp=sharing . In the final version, data will be made available directly from our system with the appropriate code.

---

### Official Review · AnonReviewer3 · 2020-10-28
**Review of the paper "Spectral Synthesis for Satellite-to-Satellite Translation"**

**Rating:** 5
**Confidence:** 5

**Review:**

=== Summary ====
This paper proposes a new method for image-to-image translation on multi-spectral imagery. The proposed method uses variational auto-encoders and generative adversarial networks to generate synthetic bands in satellite imagery.  Novelty of the proposed method is that the authors introduce a shared spectral reconstruction loss and skip connection to generate synthetic spectral bands. This allows to generate synthetic bands with higher accuracy that the original image-to image translation method.

=== Pros ====
Obtaining additional data for satellite imagery is notoriously difficult. Moreover, attempts to combine different sources of satellite imagery often fail due to the difference in resolution, band spectrum and the time, when the imagery was obtained. Therefore, all the work that aims to help solving these problems is extremely important. Authors here do not address the problem of time when the imagery was obtained (since they work with geostationary satellites), but they do address the other two: lack of data and band incompatibility. Therefore, the approach, presented in this paper, is very interesting and important application.
To our knowledge, VAE-GAN approach to satellite imagery was used before for Sentinel imagery [1], however the method, proposed here, improves this approach by introducing a new type of loss.
To summarise, the proposed method is technically correct and the application is interesting and important.

=== Cons ===
The authors claim three main contributions in the paper: “1) introduce a shared spectral reconstruction loss to a VAE-GAN architecture for synthetic band generation; 2) test our methodology on real-world scenarios; 3) present and release a test dataset of four hemispheric snapshots from three publicly available geostationary satellites for future research.”
1) The novelty of the method seems to be relatively limited and domain specific. Specifically, the original method was introduced before [2] and [3] in other applications.
Authors introduce a new type of loss to improve the performance of the model on a particular type of the input data (multispectral). This new type of loss improves the performance of the model from [3] for this particular type of input. However, the other properties of shared spectrum reconstruction loss are unclear: is it domain-specific or it can be extended to other problems?
3) Sorry, I have not found the link to the dataset, which is mentioned as the third contribution of the paper. Could you please explicitly state where you present it or remove this contribution?

=== Basis for recommendation ===
I’m borderline on this paper since the paper is very interesting and the application is important, but technical novelty is slightly limited.
Quality. The quality of the paper is very high and satisfy the standards of the ICLR publication.
Clarity. The material is presented clearly and correctly.
Originality. The paper introduces an original contribution, which was not published before (to my knowledge).
Significance. The results are important to the domain of climate-change related applications, satellite imagery processing and EO/GIS .

=== Questions to address in the rebuttal ===
1.	Please specify the novelty and importance of the proposed method. Is it possible to apply it elsewhere?
2.	Does the proposed method work with other types of imagery (other satellites/other types of sensors?) If yes, what other types of satellites/sensors can it be applied to?
3.	What are the limitations of the proposed approach?
4.	Could you please explicitly state where you present the dataset or remove this contribution?
5.	Could you explicitly state the steps for generation of new bands?
6.	For the readers, unfamiliar with satellite imagery processing, it might be interesting to understand why generation of synthetic bands is of particular importance.

=== Minor comments and additional feedback (not necessary to address in rebuttal)===
1.	It’s not very clear how authors address the problem of different band resolution.
2.	How would performance change if the tiles (or parts of tiles)  with clouds are removed from original data and there is less noise in the Sensor condition?
3.	“Shortwave infrared band cannot be inferred from others”: this is an interesting observation. It would be interesting to see in the future the accuracy with which the missing bands can be synthesised from a (random) set of bands and how accuracy increases based on the initial number of bands that were used.
4.	Can this method be applied to radar data?
References
1.	E. H. Sanchez, M. Serrurier, and M. Ortner, “Learning disentangled representations via mutual information estimation,” 2019. https://arxiv.org/abs/1912.03915
2.	A. B. L. Larsen, S. K. Sønderby, and O. Winther, “Autoencoding beyond pixels using a learned similarity metric,”  (2015) http://arxiv.org/abs/1512.09300
3.	M.-Y. Liu, T. Breuel, and J. Kautz, “Unsupervised image-to-image translation networks,” in Advances in Neural Information Processing Systems 30, (2017) http://papers.nips.cc/paper/6672-unsupervised-image-to-image-translation-networks.pdf

---

> ### Author Response · Authors · 2020-11-15
> **Response to AnonReviewer3**
>
> We thank the reviewer for the positive comments, especially regarding the importance of this work. As noted, satellite observations require synchronization in the time, spatial, and spectral dimensions for future applications. This does make obtaining satellite imagery challenging. We also appreciate sharing the first citation (Sanchez et al.) as we were not aware of this work. This solves a similar and important problem of image-to-image translation between optical sensors and segmentation tasks. We have added  (Sanchez et al.) to our paper. Since the optical sensor includes 4-bands, compared to our 16-bands, a traditional VAE-GAN architecture is sufficient. This further motivates the importance of the addition of our shared spectral reconstruction loss.
>
> Rebuttal questions:
> 1. Novelty. To better communicate the novelty in our methodology, we specifically include the UNIT [23] baseline to Table 1. The UNIT baseline is run by removing both the SSR loss and skip connection. Our method shows large improvements, emphasizing that the changes we make are not superficial.
> 2. Applications. Our approach is developed for synchronizing multi- and hyper-spectral satellite imagery with relatively similar sensors (ie. optical, near-infrared, infrared). This includes a wide range of satellite datasets and could potentially be used to create enhanced versions of legacy satellite imagery.
> 3. Limitations. We add a subsection at the end of the discussion to discuss limitations of our method.  We believe this will help with future research in the area. In particular, we note that VAEs converging to the mean may affect high precision information such as anomalies and rare events. This could have implications for science applications. In the paper, we note that performance in generating the shortwave infrared band is lower than other bands. This information is valuable for building specifications of future satellite sensors.
> 4. Yes, we have included a google drive link to share with our test set (4.9 GB, 500 GOES16 and GOES17 pairs). https://drive.google.com/file/d/1gzLcqWiKPjvzltp2nVZH6uCSIQ0G2h1u/view?usp=sharing In the final version, we will include a non-anonymous link to the dataset. Further, we have included a more detailed explanation of the data in the revisions.
> 5. We agree that the steps to generating a new band is unclear. To improve this, we have included pseudocode in the methodology section.
> 6. As discussed, in our revisions we have extended the discussion of the data and future applications.
>
> Minor Comments:
> 1. To address the problem of different sensor bands having different resolutions we use bilinear interpolation to upsample all images to the same resolution of 2km.
> 2. If the tiles (or parts of tiles) with clouds are removed from original data this may reduce the differences between the methods.  However, we would expect the directionality of the results to still be similar. The sensor condition noise is not only constrained to cloud regions.
> 3. “It would be interesting to see in the future the accuracy with which the missing bands can be synthesized from a (random) set of bands and how accuracy increases based on the initial number of bands that were used” - We agree this would be a good experiment to run. We will work on this.
> 4. Our approach may well work with multi-channel radar; however, further experimentation would need to be done.

---

### Decision · Program_Chairs · 2021-01-07
**Final Decision**

**Decision:**

Reject

**Comment:**

This paper introduces an interesting application of VAE-GAN to the problem of Spectral Synthesis across satellite observations with some additional domain specific changes (new loss, ...). The introduction of a new dataset is also very interesting and can open the door for more methodological development in the community.

While the application is original, the methodological contributions have been judged limited and domain specific by most of the reviewers. The responses from the authors was appreciated by the reviewers (especially the comparison to the UNIT baseline) but did not change their opinion about the limited methodological novelty of the approach. The AC recommends a reject but encourages the authors to resubmit it to a more applied remote sensing venue.